# Estimation of Thickness and Speed of Sound for Transverse Cortical Bone Imaging Using Phase Aberration Correction Methods: An In Silico and Ex Vivo Validation Study

**Huong Nguyen Minh [1], Marie Muller [2] and Kay Raum [1,\*]**

[1] Center for Biomedicine, Charité—Universitätsmedizin Berlin, 12203 Berlin, Germany; huong.nguyen-minh@charite.de

[2] Department of Mechanical and Aerospace Engineering, NC State University, Raleigh, NC 27695, USA; mmuller2@ncsu.edu

[\*] Correspondence: kay.raum@charite.de

**Abstract:** Delay-and-sum (DAS) beamforming of backscattered echoes is used for conventional ultrasound imaging. Although DAS beamforming is well suited for imaging in soft tissues, refraction, scattering, and absorption, porous mineralized tissues cause phase aberrations of reflected echoes and subsequent image degradation. The recently developed refraction corrected multi-focus technique uses subsequent focusing of waves at variable depths, the tracking of travel times of waves reflected from outer and inner cortical bone interfaces, the estimation of the shift needed to focus from one interface to another to determine cortical thickness ($Ct.Th$), and the speed of sound propagating in a radial bone direction ($Ct.v_{11}$). The method was validated previously in silico and ex vivo on plate shaped samples. The aim of this study was to correct phase aberration caused by bone geometry (i.e., curvature and tilt with respect to the transducer array) and intracortical pores for the multi-focus approach. The phase aberration correction methods are based on time delay estimation via bone geometry differences to flat bone plates and via the autocorrelation and cross correlation of the reflected ultrasound waves from the endosteal bone interface. We evaluate the multi-focus approach by incorporating the phase aberration correction methods by numerical simulation and one experiment on a human tibia bone, and analyze the precision and accuracy of measuring $Ct.Th$ and $Ct.v_{11}$. Site-matched reference values of the cortical thickness of the human tibia bone were obtained from high-resolution peripheral computed tomography. The phase aberration correction methods resulted in a more precise (coefficient of variation of 5.7%) and accurate (root mean square error of 6.3%) estimation of $Ct.Th$, and a more precise (9.8%) and accurate (3.4%) $Ct.v_{11}$ estimation, than without any phase aberration correction. The developed multi-focus method including phase aberration corrections provides local estimations of both cortical thickness and sound velocity and is proposed as a biomarker of cortical bone quality with high clinical potential for the prevention of osteoporotic fractures.

**Keywords:** medical beamforming; phase aberration correction; medical tissue characterization; pulse-echo ultrasound; medical signal and image processing

## 1. Introduction

The current standard method for bone strength assessment and fracture risk prediction is based on areal bone mineral density (*aBMD*) measured by dual-energy absorptiometry (DXA) [1]. Although *aBMD* is an important biomarker of bone quality, additional bone factors, including macro- and micro-structural bone parameters, as well as viscoelastic properties, are known to determine individual bone strength; therefore, to quantify these parameters for bone assessment, quantitative ultrasound (QUS) methods have been introduced as nonionizing alternatives. Early bone QUS technologies used dedicated

hardware to measure acoustic properties, such as the speed of sound (*SOS*) and broadband ultrasound attenuation (*BUA*), at anatomical sites that contain mostly trabecular bone, such as the heel [2]. More recent QUS devices are aimed at imaging bone by using dedicated hardware electronics and ultrasound probes. An example of this by Lasaygues et al. developed ultrasonic image reconstruction methods to image the cortical diaphysis of long bones using quantitative ultrasonic tomography [3,4]. Another tomographic approach to image long bones is based on full-waveform inversion [5]. Additionally, Li et al. used Split-Step Fourier imaging to image bone fractures and to monitor bone healing [6]. Furthermore, a Born-based inversion method has been implemented on an ultrasonic wavefield imaging technique to reconstruct internal structures of long bones [7]. Limitations of these studies were that either the speed of sound or the thickness needed to be assumed a priori. Axial transmission devices can retrieve cortical parameters (i.e., porosity, thickness, and speed of sound), by measuring the propagating velocity of dispersive guided waves [8–12]; however, this technique is challenged by large soft tissue thickness, irregular bone shapes, and it does not provide direct image guidance.

A few recent technologies utilize sophisticated array-based pulse-echo imaging technology to estimate *BMD* in trabecular bones at major fracture sites (i.e., spine and proximal femur [13]), or to measure structural and material properties in the cortical bone (i.e., tibia and radius) [14,15]. Most medical ultrasound scanners on the market implement the standard delay-and-sum (DAS) beamforming method to reconstruct the brightness mode (hereinafter called B-mode) images. This technique uses a transducer array to transmit and receive focused ultrasound signals inside the body. Conventionally, the reconstruction of B-mode images using DAS is done by adding time specific delays to the individual ultrasound signals which are received at each element of the receiver array before summing all signals to create a beamformed received signal; therefore, the sensitivity of the beamformed signal can be maximized to a certain depth and direction. In medical ultrasound scanners, transmit and receive focusing is performed by assuming a constant speed of sound of soft tissue (1540 m/s) along the entire sound propagation path. This approximation provides satisfactory image quality for most soft tissues, because the true velocities only vary within 10% when compared with the assumed value [16]; however, this is not the case for mineralized tissues, such as cortical bones. The radial speed of sound in cortical bone is between the range of 2800 to 3500 m/s [17], which results in a substantial refraction at the soft tissue and cortical bone interface. In case of a wrong assumption on the constant sound velocity, the delay estimation, which is necessary to focus on a particular image location, is incorrect, subsequently leading to a phase-distorted DAS signal. As a result, in a conventional B-mode image reconstructed by medical ultrasound scanners, the internal bone structures appear blurred or cannot be reconstructed at all. Aside from radiofrequency echographic multispectrometry (REMS) technology [18], this conventional DAS beamforming is currently used for bone strength assessment and fracture risk prediction. There have been efforts to overcome this false assumption of a constant speed of sound in cortical bone. Renaud et al. [14] proposed the first in vivo image reconstruction of cortical bone using a conventional medical ultrasound scanner and seismic image reconstruction. This reconstruction method provides local estimations of *Ct.Th* and anisotropic sound velocity.

Consequently, the need for further methods brought about the multi-focus (MF) imaging technique, that was developed by our group to measure cortical thickness (*Ct.Th*) and the compressional sound velocity propagating in the radial bone direction (*Ct.*$v_{11}$) [19]. Our method aims at imaging cortical bone at the central anteromedial tibia. This anatomical site is of clinical interest, as it is easy to access, and is composed of a thick and regular cortical bone shell. Alterations, such as reduced cortical thickness and the occurrence of large intracortical pores, have shown to be associated with reduced hip strength [20] and increased fracture risk [21]. The ultrasonic speed of sound in cortical bone has been proposed as a biomarker of bone quality since the 1970s [22], and is related to bone density and elastic constants, which are correlated to bone quality and fracture risk [23].

The MF method is based upon the consecutive focusing of ultrasound waves at varying depths, followed by the retrieval of focus locations and pulse-travel times of signals reflected from the periosteal (frontside) and endosteal (backside) cortical bone interfaces using conventional DAS beamforming (Figure 1). So far, the MF method has been validated on plates with constant thickness, positioned parallel to the probe array; however, it is important to extend this, as typical human cortical bones exhibit curvatures at their periosteal and endosteal interfaces. These curvatures introduce a distortion of the propagating wavefronts and the round-trip travel time, resulting in phase distorted beamformed signals. The objective of this research is to analyze the effect of the phase aberration caused by (a) bone curvature, (b) bone tilt with respect to the beam axis, and (c) material inhomogeneities due to the presence of cortical pores on the estimations of *Ct.Th* and *Ct.$v_{11}$*. The phase aberration from bone surface curvature leads to a different round-trip travel time when compared with a flat plate bone model (surface time shift, ST), and was corrected using the concept of refraction compensation proposed by Yasuda et al. [24]. Bone tilt, with respect to the beam axis of the transducer array, shows orientation dependence of the received echoes compared with a flat bone interface. Additional time shifts caused by the orientation dependence of received echoes were corrected using autocorrelation analysis (ACF). Differences in the round travel time of the received echoes based on the interaction of ultrasound wave refractions with cortical pores were determined using cross correlation analysis (CC). We show the need to incorporate three phase-aberration correction (PAC) methods for non-plate shaped bone structures by means of numerical finite-difference time-domain (FDTD) simulation models, as well as measurements on a human tibia bone. Precision and accuracy values of estimated *Ct.Th* and *Ct.$v_{11}$* with and without corrections were compared.

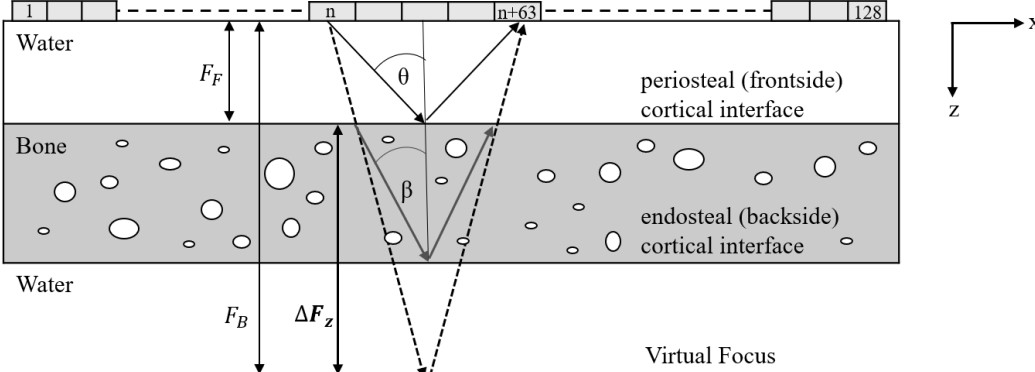

**Figure 1.** Schematic representation of the multi-focus measurement in the radial direction (*x, z*) of a long bone. The transducer is placed 15 mm above the sample. Focused sound beams are emitted using a 64-element sub-aperture of a 128-element linear array. The focus is shifted from a depth above the periosteal cortical interface to a depth below the endosteal cortical interface by gradually decreasing the semi-aperture angle *θ* of the transmit beam. Refraction at the periosteal interface changes the direction of the transmitted waves and results in a shift of the focus depth inside the bone. *ΔFz* is the focus depth shift required to focus from the periosteal (frontside *$F_F$*) to the endosteal (backside *$F_B$*) interface. In addition to scanning the focus depth, sub-aperture is scanned in the *x*-direction along the transducer array (adapted from [19] under the Creative Commons Attribution 4.0 license).

## 2. Materials and Methods

An overview for all used abbreviations is summarized in Table 1.

**Table 1.** List of abbreviations.

| Abbreviation | Description |
|---|---|
| $Ct.Th$ | Cortical thickness |
| $Ct.v_{11}$ | Cortical compressional sound velocity propagating in the radial bone direction |
| $v_{H2O}$ | Speed of sound in water |
| $dx$ | Lateral shift of center of mass of curved bone plate model relative to beam axis |
| $r$ | Bone plate curvature radius |
| $Ct.Po$ | Cortical porosity |
| $F_z$ | Focus depth in $z$-direction |
| $H_F(F_z)$ | Amplitude of Hilbert-transformed envelope signal of beamformed frontside reflection at focus depth $F_z$ |
| $H_B(F_z)$ | Amplitude of Hilbert-transformed envelope signal of beamformed backside reflection at focus depth $F_z$ |
| $FB$ | Front- and backside reflection |
| $\Delta TOF$ | Shift in time-of-flight between peak position of $H_F(F_z)$ and $H_B(F_z)$ |
| $\Delta F_z$ | Shift in focus depth between peak position of $H_F(F_z)$ and $H_B(F_z)$ |
| $F_{z,B}$ | Confocal focus depth position of backside reflection |
| $\theta$ | Semi-aperture angle of transmit and receive beams |
| $k_{eff}$ | Correction factor $k_{eff}$ for effective aperture $k_{eff}\theta$ |
| $\theta_{crit}$ | Critical angle based on Snell's law |
| $\Delta\theta$ | Difference of the semi-aperture angle to the critical angle |
| $Tx_i, Rx_i$ | Transmit or receive channel number |
| $Rx_{ref}$ | Reference receive channel with maximum amplitude at envelope signal of pre-beamformed backside reflection |
| $V_{gb}$ | Gated pre-beamformed backside reflection signals |
| $V_{ACF}$ | Signal after using autocorrelation function (ACF) |
| $|V_{ACF}|$ | Magnitude of the ACF signal |
| $\alpha_{ACF}$ | Inclination angle of the fitted ellipsoid on $V_{ACF}$ to the major semi-axis |
| $\Delta t_{ACF}$ | Time shift correction based on $\alpha_{ACF}$ |

*2.1. Numerical Ultrasound Propagation Model*

Ultrasound wave propagation in bone and water was simulated using a 2D finite-difference time-domain (FDTD) code (Simsonic, www.simsonic.fr, accessed on 10 March 2022) [25]. The simulation model considers elastic wave propagation including mode conversion, multiple scattering, frequency-independent absorption, refraction, and diffraction. A convergence study, as described in [19], provided stable results at grid sizes of 7 µm and time steps of 0.93 ns. Table 2 shows the material properties used for the models in this study. Material properties were used from an ex vivo study [26] and a previous acoustic microscopy study in a human femoral cortical bone [27]. All bone models were simulated as hollow cylinders immersed in water. The cylinders were defined by an outer curvature radius $r$ and a wall thickness $d$. All bone models were placed 15 mm below a linear array with 64 transmitter and receiver elements (element and pitch sizes: 0.3 mm); therefore, the models assumed the sound propagation in the transverse image plane (i.e., perpendicular to the bone's long axis, at the antero-medial midshaft of a tibia, where the outer bone surface is flat or slightly curved and the sound velocity of the tissue matrix can be assumed to be isotropic in the simulation plane). The transducer elements emitted broadband pulses with a center frequency of 4 MHz and a -6-dB bandwidth of 60%. Phase delays were applied to focus the transmit beam consecutively, at depths ranging from 13 mm to 40 mm, with an increment of 1 mm. The signals received by all elements were captured and downsampled to a sampling rate of 80 MHz for further processing. The sufficient aperture size of 64 was chosen based on a side study, which can be found in Appendix A.

**Table 2.** Tissue material properties of bone and pores used for the numerical model. Mass density $\rho$, and $c_{ij}$ (i.e., the coefficients of a transverse isotropic stiffness tensor were taken from [27] and the absorption value $\alpha$ was obtained from [26]) (adapted from [19] under the Creative Commons Attribution 4.0 license).

|  | **Bone** | **Pores/Water** |
|---|---|---|
| $\rho$ [g/cm³] | 1.93 | 1.00 |
| $c_{11}$ [GPa] | 23.7 | 2.25 |
| $c_{22}$ [GPa] | 23.7 | 2.25 |
| $c_{12}$ [GPa] | 9.5 | 2.25 |
| $c_{66}$ [GPa] | 6.6 | 0 |
| $v_{11}$ [m/s] | 3504 | 1500 |
| $\alpha$ [dB/mm] | 2.1 | 0.002 |

### 2.1.1. Reference Bone Model: Flat Bone Plate

The reference model consisted of a 4 mm thick bone plate ($Ct.Th^{Ref}$ = 4 mm) without pores. The material properties of the homogenous bone material results in a reference speed of sound of $Ct.v_{11}^{Ref}$ = 3504 m/s. The curvature radius of $r$ = 10 m was used to simulate a flat bone plate (hereinafter simply called 'flat bone plate'). The radius of 10 m was deemed sufficiently large to exhibit a negligible curvature within the simulation region (Figure 2a).

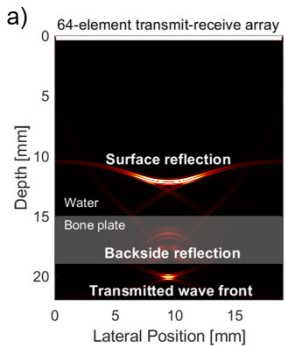 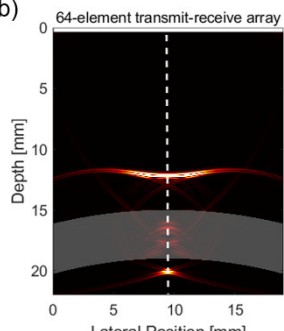 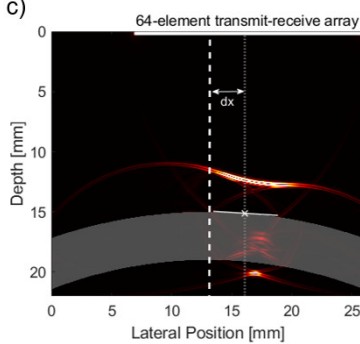 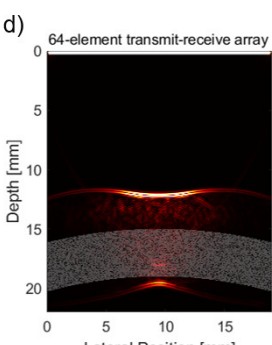

**Figure 2.** (**a**) Snapshot of the flat bone plate simulation model at 13.5 μs. Transmitted and reflected wavefronts of a beam generated with a 64-element aperture and focused to a depth of 25 mm can be seen. (**b**) Snapshot of the curved plate model with a curvature radius of 40 mm. (**c**) Snapshot of the curved plate model *r40dx3.11* with vertical bone symmetry axes being marked by a white dashed line, beam axis is shown by the grey dashed line, and the cross point of the beam axis with the frontside surface is marked as white. (**d**) Snapshot of *r40dx0Po16* with a cortical porosity of 16% and pore diameter 60 μm at focal depth of 25 mm.

### 2.1.2. Bone Curvature

To investigate the effect of bone curvature, curved bone plate models were simulated and compared with the flat bone plate model (Figure 2b). Five curved bone models with radii of $r$ = 60 mm, 50 mm, 40 mm, 30 mm, and 20 mm were simulated. The radius range was defined based on a previous study, in which human tibia midshaft bones of 55 post-menopausal women were measured by means of high-resolution peripheral computed tomography (HR-pQCT) [15]. In that study, the anteromedial tibia midshaft region had been chosen as the ultrasound measurement site due to the small amount of overlying soft tissue and the small curvature of the bone surface compared with other tibia regions. To estimate the curvature radius, circular fits were performed on the central anteromedial tibia region. Tibia bone curvature radii were found to be in the range between 12.6 mm and 68.8 mm with a mean radius of 30.3 mm. Three examples of the circular fits on the

HR-pQCT scans are shown in Figure A2 of Appendix B, where the subjects with a minimum (Figure A2a), mean (Figure A2b), and maximum curvature radius (Figure A2c) were selected.

### 2.1.3. Bone Tilt

To study the effect of the angle of incidence, a bone surface tilt was incorporated by shifting the lateral position of the transducer array by *dx* (Figure 2c). The bone surface tilt was defined as the angle between the normal vector of the periosteal bone surface and the beam axis at their crossing point.

### 2.1.4. Material Inhomogeneity: Cortical Pores

To study the effect of material inhomogeneity, cortical pores were included in the curved bone plate models (Figure 2d). Previous ex vivo studies in human cortical bone reported cortical porosity (*Ct.Po*) and cortical pore diameter (*Ct.Po.Dm*) values between 2% and 22% and 7 and 95 µm, respectively [28–30]. Cortical pores were defined as circular pores with *Ct.Po.Dm* = 60 µm and varying pore densities, resulting in models with *Ct.Po* values ranging from 0% to 20% with an increment of 2%.

For the simulation models with cortical pores, transmission measurements were performed to calculate the reference speed of sound $Ct.v_{11}^{Ref}$. An unfocused single-element transducer with a width of 0.3 mm emitted ultrasound waves with a center frequency of 4 MHz and a -6-dB bandwidth of 60%. The unfocused ultrasound wave traveled though the bone and the transmitted ultrasound wave was captured by a single element detector with a width of 0.3 mm, which was placed below the bone. The transducer and detector were placed at the beam axis of the reference MF simulation. In addition, a simulation was performed with the same configuration without the bone to measure the reference signal transmitted though water. The time-of-flight of the ultrasound wave transmitted through water $TOF_{H20}$ and bone $TOF_{bone}$ was defined at the time of the maximum of the signal envelope. The $Ct.v_{11}^{Ref}$ of the bone models with pores were calculated using the following equation from [31].

$$Ct.v_{11}^{Ref} = \frac{Ct.Th_{bone}}{\frac{Ct.Th_{bone}}{v_{H_2O}} + \left(TOF_{bone} - TOF_{H_2O}\right)}, \tag{1}$$

with *Ct.Th*<sub>bone</sub> = 4 mm.

### *2.2. Ex Vivo Measurement on a Human Tibia Bone*

One left tibia bone from a human cadaver (female, age 85) was used for the ex vivo validation. The bone sample was received without the soft tissue and distal end (cut off at approximately 50%). The sample was collected by the institute of Anatomy, University of Lübeck, Germany, in accordance with the German law "Gesetz über das Leichen-, Bestattungs- und Friedhofswesen des Landes Schleswig-Holstein II Abschnitt, §9 Leichenöffnung, anatomisch", from 2 April 2005. A 30 mm disk was cut from the tibia midshaft using a band saw (EXACT GmbH, Remscheid, Germany). A HR-pQCT scan was performed (XtremeCT II, Scano Medical AG, Bassersdorf, Switzerland) with a total scan length of 10.2 mm in the axial direction and an isotropic voxel size of 60.7 µm. Cortical thickness at the anteromedial tibia section was extracted using a custom protocol adapted from Iori et al. [32] and used as reference value. Cortical porosity was calculated from the HR-pQCT scan using the algorithm proposed by Burghardt et al. [33]. A site-matched multi-focus measurement was performed using a medical scanner SonixTouch equipped with a 3D linear array transducer 4DL14-5/38 (consisting of a 1D 128 element array) and a SonixDAQ single-channel data acquisition system (Ultrasonix, Richmond, BC, Canada). The SonixDAQ allows the pre-beamformed single-channel radio frequency (RF) data acquisition of all channels without any signal processing. The sample was immersed in water and the transducer array was positioned perpendicular to the bone's long axis. The

multi-focus measurement sequence consisted of a series of conventional B-mode imaging sequences with 128 lateral scan positions. At each scan position, sound waves were focused on the radial bone where the direction was into the tibia sample using a 64-element transmit aperture. Subsequent B-mode images were acquired using 17 gradually increasing focus depths (starting from 14 mm with a step size of 2 mm). The transducer elements were excited with a "+−" signal at a system transmit frequency of 4 MHz to optimize the penetration depth. Single-channel RF data were captured with all 128 array elements at a sampling rate of 40 MHz with a 12-bit resolution.

### *2.3. Signal Processing*

#### 2.3.1. Reference Bone Model: Flat Bone Plate

Details of the multi-focus signal processing steps have been described previously [19]. From the delay and beamformed (DAS) Hilbert-transformed envelope signal, the amplitudes [$H_F(F_z)$ and $H_B(F_z)$] and pulse travel times [$TOF_F(F_z)$ and $TOF_B(F_z)$] of the signals reflected from the front- and backsides of the plate were tracked for each beam focus position $F_z$. The time-of-flight difference between front- and backside reflections was defined as $\Delta TOF = TOF_B(F_z) - TOF_F(F_z)$. Spline interpolation was used to estimate $H_F(F_z)$ and $H_B(F_z)$ at an $F_z$ increment of 0.1 mm. The interpolated data, and the front- and backside focus positions $F_F$ and $F_B$, respectively, were retrieved from the peak positions of $H_F(F_z)$ and $H_B(F_z)$, and $\Delta F_z$ (i.e., the shift needed to focus either on the front- or backside of the plate, and to estimate the time delay between front- and backside reflections $\Delta TOF$). $Ct.Th$ and $Ct.v_{11}$ were estimated using Equation (3) in [19] with sound velocity in water $v_{H20}$:

$$Ct.Th = \frac{\Delta F_z}{0.5 \cdot \frac{Ct.v_{11}}{v_{H_2O}}\left(1 - \frac{Ct.v_{11}^2}{v_{H_2O}^2}\right)\cdot\left(1 - \cos\left(k_{eff}\theta\right)\right) - \frac{Ct.v_{11}}{v_{H_2O}}}, \tag{2}$$

where $\theta$ is the semi-aperture angle of the transmitting and receiving beams, and $k_{eff}$ is an effective aperture contributing to the beam focusing on the backside of the plate. The effective aperture accounts for the increased conversion of compressional waves into shear waves with increasing inclination angles and the absence of compressional wave transmission into the bone tissue for inclination angles larger than the critical angle $\theta_{crit}$ [19]:

$$\theta_{crit} = \sin^{-1}\left(\frac{v_{H_2O}}{Ct.v_{11}}\right). \tag{3}$$

In contrast to our previous study [19], we have used an aperture size of 64 elements and adjusted the factor to estimate the effective aperture $k_{eff}$ from 0.1 to 0.122:

$$k_{eff} = \begin{pmatrix} 1 & \text{if } \theta < \theta_{crit} - 10° \\ 0.122 \cdot \Delta\theta & \text{if } \theta > \theta_{crit} - 10° \end{pmatrix}. \tag{4}$$

More details on the estimation for $k_{eff}$ can be found in Appendix A.

#### 2.3.2. Phase Aberration Correction

Phase aberrations caused by bone curvature, bone tilt and material inhomogeneities are corrected for signals reflected from the backside cortical bone interface. Three phase-aberration correction (PAC) methods are used: (1) The curved bone surface geometry results in different round-trip travel times compared with the flat bone model. A time-shift correction based on the periosteal bone surface geometry (hereinafter called 'surface time correction' ST), was used to correct for the additional ultrasound wave propagation paths in the water due to the bone curvature. The ST correction used the concept of refraction compensation proposed by Yasuda et al. [24]. Further details are summarized in Figure A3a in Appendix C. (2) For tilted bone models, the reflected wavefront exhibits a tilt with respect to the beam axis (Figure 2c). To correct the phase aberration caused by surface inclination, an autocorrelation function (ACF) analysis on the reflected backside echoes was performed (Figure A3b–d in Appendix C). (3) Local variations of the sound velocity

caused by material inhomogeneities lead to small fluctuations of the transit time measured at individual receiver elements and subsequently to a distortion of the summed signal; therefore, the following method was used to estimate the backside focus depth. The arrival times for all receiver elements was estimated using a cross-correlation (CC) method. The receiver channel that measured the highest signal amplitude was used as the reference signal. The inter-element arrival times exhibit either a concave, flat, or a convex shape, depending on the distance of the beam focus relative to the backside bone interface. A second-order polynomial was fitted to the inter-element arrival times, and the confocal focus depth was determined by finding the zero-crossing point of the second order fit coefficients (Figure A3e,f in Appendix C). This zero-crossing point was used to determine $\Delta F_z$, and to estimate $Ct.Th^{MF}$ and $Ct.v_{11}^{MF}$ using Equation (2).

### 2.4. Statistics

For each model, the retrieved $Ct.Th^{MF}$ and $Ct.v_{11}^{MF}$ values were compared with the reference $Ct.Th^{Ref} = 4$ mm and $Ct.v_{11}^{Ref}$. Simulation models without cortical pores had the reference speed of sound of $Ct.v_{11}^{Ref} = 3504$ m/s. The bone models with cortical pores $Ct.v_{11}^{Ref}$ were extracted from the transmission measurements. Pearson linear regression analysis was performed to compare the parameters obtained using the multi-focus method with reference values. For all models with a 64-element aperture, the relative error (RE), precision, and accuracy values for each PAC method were determined and compared with the values without any PAC. Precision was defined as the coefficient of the variation of the difference between the predicted $Ct.Th^{MF}$, $Ct.v_{11}^{MF}$ and the reference values for $Ct.Th^{Ref}$, $Ct.v_{11}^{Ref}$. Accuracy was determined by means of the root mean square error (RMSE) compared with the reference values. All analyses were performed using MATLAB R2019b, including the Signal Processing, Curve Fitting, and Statistics Toolboxes (The Mathworks, Natick, MA, USA).

## 3. Results

### 3.1. Numerical Simulations

A total of 22 bone models were simulated (Table A2 in Appendix D). The reference sound velocities $Ct.v_{11}^{Ref}$ of the porous models, as determined by transmission simulations, are summarized in Table A3 in Appendix D. The estimated $Ct.Th^{MF}$ and $Ct.v_{11}^{MF}$ values for all models and the relative errors are summarized in Table A4 in Appendix D. Without PAC, all deviations from the ideal flat plate geometry led to deteriorations of precision and accuracy. In most situations, PAC improved both the precision and accuracy (Tables 3 and 4), which will be described in more detail in the following sections.

**Table 3.** Precision of $Ct.Th^{MF}$ and $Ct.v_{11}^{MF}$ after each PAC method.

|  | Model | No PAC | ST | ST + ACF | ST + ACF + CC |
|---|---|---|---|---|---|
| | Curved bone plate | 4.3% | 1.7% | 1.4% | 2.0% |
| $Ct.Th^{MF}$ | Curved tilt bone plate | 2.3% | 7.3% | 4.3% (4.2%) * | 4.3% (1.1%) * |
| | Material inhomogeneity | 18.5% | 1.4% | 4.7% | 7.2% (1.9%) ** |
| | Curved bone plate | 4.3% | 1.6% | 1.4% | 2.0% |
| $Ct.v_{11}^{MF}$ | Curved tilt bone plate | 2.3% | 7.1% | 4.2% (2.5%) * | 4.3% (0.8%) * |
| | Material inhomogeneity | 15.9% | 7.9% | 7.9% | 8.2% (7.9%) ** |

\* Exclusion of bone models with tilt angles over 7°. ** Exclusion of bone model with porosity 20%.

**Table 4.** Accuracy of $Ct.Th^{MF}$ and $Ct.v_{11}^{MF}$ after each PAC method.

|  | Model | No PAC | ST | ST + ACF | ST + ACF + CC |
|---|---|---|---|---|---|
| | Curved bone plate | 10.2% | 2.2% | 1.9% | 1.8% |
| $Ct.Th^{MF}$ | Curved tilt bone plate | 9.6% | 10.3% | 5.2% (1.3%) * | 5.2% (1.2%) * |
| | Material inhomogeneity | 23.2% | 14.6% | 6.3% | 8.3% (3.5%) ** |

| | | | | | |
|---|---|---|---|---|---|
| $Ct.v_{11}^{MF}$ | Curved bone plate | 10.4% | 2.4% | 2.1% | 1.9% |
| | Curved tilt bone plate | 9.8% | 10.1% | 5.1% (1.4%) * | 5.1% (1.2%) * |
| | Material inhomogeneity | 25.3% | 3.4% | 3.4% | 2.8% (3.0%) ** |

\* Exclusion of bone models with tilt angles over 7°. ** Exclusion of bone model with porosity 20%.

### 3.1.1. Effect of Bone Curvature

All three PAC methods showed improvements of $Ct.Th^{MF}$ and $Ct.v_{11}^{MF}$ estimations with respect to precision and accuracy (Tables 3 and 4). Although the ST correction alone showed the strongest improvement, the combination of all three PAC only yielded slight further improvements.

### 3.1.2. Effect of Bone Tilt

To correct for the bone tilt, using ST correction was not sufficient, and it even degraded accuracy and precision values (Tables 3 and 4). The wavefront inclination caused by the tilted surface was effectively corrected using the ACF; however, for bone models with tilt angles above 7°, the CC correction method failed, because no zero-crossing point for the estimation of confocal focus depth could be retrieved (Figure A4c in Appendix D). After excluding these models, precision values for $Ct.Th^{MF}$ and $Ct.v_{11}^{MF}$ were 1.1% and 0.8%, respectively, and accuracy values were 1.2% for both the $Ct.Th^{MF}$ and $Ct.v_{11}^{MF}$ estimations.

### 3.1.3. Effect of Material Inhomogeneities

The presence of pores strongly degraded accuracy and precision values without PAC. The ST correction strongly improved precision and accuracy. Additional ACF correction had no effect in the evaluated simulations, because all porous bone models were modeled without a tilt. The CC further improved precision and accuracy values for $Ct.Th^{MF}$ and $Ct.v_{11}^{MF}$. For the bone model with the highest porosity value of 20%, all PAC methods did not result in a precise and accurate estimation of $Ct.Th^{MF}$ and $Ct.v_{11}^{MF}$ (Table A4 and Figure A5 in Appendix D).

### 3.1.4. Overall Effect of PAC

Table 5 shows the precision and accuracy values for all 22 simulation models. Note that precision values are defined as the coefficient of variation of the difference between the estimated and reference value of the flat bone plate model and accuracy is defined as RMSE as a percentage. That means the smaller the precision and accuracy value, the more precise and accurate the parameter estimation is with respect to the reference value. Overall, the combination of the three PAC methods results in an improved precision and accuracy estimation $Ct.Th^{MF}$ and $Ct.v_{11}^{MF}$. Precision values for $Ct.Th^{MF}$ and $Ct.v_{11}^{MF}$ were 5.7% and 9.8%, respectively. Accuracy values for $Ct.Th^{MF}$ and $Ct.v_{11}^{MF}$ were 6.3% and 3.4%, respectively.

**Table 5.** Precision and accuracy of $Ct.Th^{MF}$ and $Ct.v_{11}^{MF}$ for each PAC method for all 64-element aperture models. Reference thickness for all models is $Ct.Th^{Ref}$ = 4 mm. Models without cortical pores have $Ct.v_{11}^{Ref}$ = 3504 m/s. Reference values $Ct.v_{11}^{Ref}$ for models including cortical pores can be found in Table A3 in Appendix D.

| | Correction | Precision | Accuracy |
|---|---|---|---|
| $Ct.Th^{MF}$ | No | 17.4% | 17.1% |
| | ST | 10.3% | 11.2% |
| | ST + ACF | 4.1% | 5.2% |
| | ST + ACF + CC | 5.7% (2.1%) * | 6.3% (2.6%) * |
| $Ct.v_{11}^{MF}$ | No | 11.6% | 18.5% |
| | ST | 9.5% | 5.9% |

| | | |
|---|---|---|
| ST + ACF | 9.5% | 3.7% |
| ST + ACF + CC | 9.8% (9.6%) * | 3.4% (2.3%) * |

\* Exclusion of bone models with tilt angles over 7° and/or porosity above 20%.

### 3.2. Ex-Vivo Multi-Focus Measurement

Reference cortical thickness and porosity values of the human tibia bone at the central anteromedial part were found to be $Ct.Th^{Ref}$ = (2.65 ± 0.61) mm and 15.1%, respectively, using HR-pQCT. For the MF measurement, $Ct.Th^{MF}$ and $Ct.v_{11}^{MF}$ were determined in a manually selected region of interest (Figure 3a green ROI). The maximum amplitudes over all focus depths with and without PAC methods (ST + ACF + CC) are shown in Figure 3a,b. The endosteal surface of the human tibia sample is more blurred in the maximum projection image without PAC methods (Figure 3a). Without PAC, $Ct.Th^{MF}$ and $Ct.v_{11}^{MF}$ could be retrieved at 14 lateral scan positions, whereas with PAC, cortical parameter estimations were achieved at 29 scan positions. The mean and standard deviation of $Ct.Th^{MF}$ without PAC was (2.39 ± 0.25) mm, which was significantly different from the reference value. In contrast, the estimation of $Ct.Th^{MF}$ with PAC of (2.71 ± 0.22) mm was not significantly different from the reference value. The estimated cortical speed without and with PAC were (2870 ± 95) m/s and (2857 ± 52) m/s, respectively.

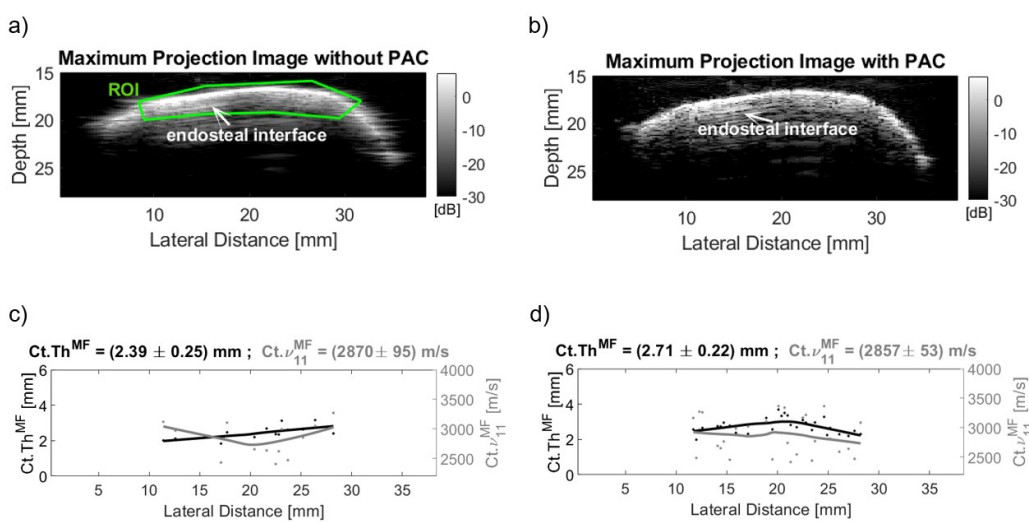

**Figure 3.** (**a**) Maximum projection B-mode image of human tibia bone at the central anteromedial region. The image was reconstructed and spatially compounded (by means of maximum projection) from measurements at all focus depths using conventional DAS beamforming. The range of interest (ROI) was manually selected (green lines). (**b**) Maximum projection B-mode image reconstructed from all focus depths with PAC (ST, ACF, CC). (**c**) Representative plots of $Ct.Th^{MF}(x_i)$ and $Ct.v_{11}^{MF}$ $(x_i)$ without PAC and (**d**) with PAC. The dots indicate the estimations for each individual lateral scan position $x_i$, and the straight lines are the estimations using a moving average filter. Means and standard deviations were determined from smoothed data. The number of individual scan positions contributing to the parameter estimations in (**c**,**d**) were 14 and 29, respectively.

### 4. Discussion

In this study, we have extended the estimations of thickness and speed of sound in cortical bone in a transverse plane using the multi-focus approach to realistic bone geometries. For this, several phase aberration corrections were proposed. The effects of bone curvature, surface inclination relative to the beam axis, and the presence of intracortical pores' parameter estimations were analyzed.

*4.1. Numerical Simulation*

### 4.1.1. Effect of Bone Curvature

For curved bone models positioned parallel to the probe array (without bone tilt), the ST correction was sufficient and corrected the additional geometrical time shifts for curved bone interfaces compared to a flat bone plate. For the correction, it was assumed that ultrasound waves propagate in a straight direction, as described in ray theory [34].

### 4.1.2. Effect of Bone Tilt

Additional phase aberration corrections on the backside echoes were necessary for the curved models with bone tilt relative to the beam axis, to correct the orientation dependence of the reflected wavefront. Here, autocorrelation function was used on the backside echoes to estimate the inclination of the backside echoes; therefore, an ellipsoid was fitted on the magnitude of the backside signals after autocorrelation analysis.

### 4.1.3. Effect of Material Inhomogeneities

Cortical pores result in scattering and subsequent diffusion of the ultrasound waves. This causes local fluctuations of the arrival time of the received backside echoes compared with the reference flat bone plate model. The backside confocal depth, which is required for the simultaneous estimation of both thickness and sound velocity, has been estimated in our previous work by detecting the peak position of the DAS beamformed backside echoes with respect to the focusing depth [19]. Phase aberration induced by cortical pores cause a decrease in the intensity of the beamformed signal. With increasing porosity, the confocal peak arising from the backside reflection becomes less sharp and the peak position is harder to detect; therefore, we have developed another method to extract the confocal backside position by analyzing the curvature of the backside echo wavefront prior to the summation of all receive channels at each focus depth. The curvature of the wavefront was extracted by analyzing the cross correlation of the backside echoes relative to the backside echo with the highest signal amplitude. The change of the wavefront curvature from a convex shape (negative curvature) to a concave shape (positive curvature) was used to extract the focus position. The zero-crossing point was calculated using a linear fit of the retrieved curvature values over the focus depth. Incorporating cross correlation analysis prior to the summation of the beamformed signals improved the accuracy of the estimation of the backside confocal position, as well as precision and accuracy in simulations including pores (Table 5). Moreover, this method improved the backside signal detection rate and the accuracy of the estimation of cortical thickness in the ex-vivo measurement.

### 4.1.4. Combination of Phase Aberration Methods

For the transition to in-vivo applications of the multi-focus method, the combination of all three PAC methods is necessary, because all the investigated deviations from an ideal flat homogenous plate are present in real cortical bone. Overall, the combination of the three PAC showed a strong improvement of precision and accuracy values for cortical thickness and speed of sound estimations than when compared to the values without PAC.

*4.2. Ex Vivo Measurement*

The endosteal surface of the human tibia sample was tracked with and without PAC methods; however, more endosteal surface locations were retrieved when PAC was used. The cortical thickness measured by ultrasound was consistent with the reference value measured by HR-pQCT. The cortical sound velocity of (2857 ± 53) m/s was in the range of the cortical speed of sound values typically found in human cortical bone [17]. Our previous study showed a dependency of cortical speed of sound on cortical porosity (*Po*) $Ct.v_{11}{}^{fit}$

= $0.39 \cdot Po^2 - 51.4 \cdot Po + 3485$ [m/s], Figure 6a in [19]). By inserting the reference cortical porosity value of 15.1% obtained from HRpQCT into this equation a speed of sound value of $Ct.v_{11}{}^{fit}$ = 2804 m/s was determined for the human tibia sample. The MF-based estimation was in the range of the expected speed of sound value; however, this observation needs to be confirmed in a larger sample size. In conclusion, the ex vivo measurement on a human tibia sample suggests the ability to measure cortical thickness and speed of sound using the MF approach by incorporating PAC methods.

### 4.3. Transition to In Vivo Application

Cortical bone has been proposed as significant predictor of a bone's mechanical strength because mechanical force given to a bone is carried primally by cortical bone [35]. Clinical studies showed an improvement of fracture prediction by measuring cortical thickness [36–38]. HR-pQCT is the most precise modality to measure cortical thickness at the tibia with a precision of 1.6% [38]. Our study showed a thickness precision estimation of 5.7%. We expect that the clinical precision of the MF approach could be larger than for controlled simulations; however, HR-pQCT uses ionizing radiation and is extremely expensive compared with ultrasound imaging. Wydra et al. [39] proposed a similar refraction measurement method and reported precision values for $Ct.Th$ of 8.5% for measurements on porous plate-shaped skull bone phantoms. In contrast, our study considered bone curvature and bone tilt with a better precision value of 5.67%, which can be attributed to the PAC methods, the use of a higher frequency (4 vs. 2.25 MHz), and the consideration of an effective aperture [19].

In addition to cortical thickness, ultrasonic wave-speed in cortical bone has been proposed as a biomarker for bone quality [10,40–42]. Bidirectional axial transmission techniques use a probe with several ultrasonic transmitters and receivers to measure waves traveling in the longitudinal direction of long bones. An in vivo study by Minonzio et al. used a bidirectional axial transmission technique (BDAT) to estimate the cortical thickness and porosity, and they reported those parameters as suitable biomarkers for fracture discrimination in postmenopausal women [43]. The QUS device Bindex® calculates the apparent cortical thickness at the distal radius and tibia using BDAT and reported the correlation with BMD (r ≥ 0.71, $p < 0.001$, $0.20 < R^2 < 0.55$) [44]. Talmant et al. [41] showed that the velocity of the first arriving signal (vFAS) is a significant biomarker for fracture discrimination and to predict fracture risk in vivo. Inter-operator precision (repeated measurements by different operators) for FAS velocities were reported at ~7%, respectively. In our study we report the precision value for different simulation models (precision of radial cortical speed of sound was 9.8%), which have been simulated only once. Compared with axial transmission techniques, the multi-focus measurement estimates cortical thickness and speed of sound within the imagined plane and provides image guidance.

Another approach to measure $Ct.Th$ and $Ct.v_{11}$ using corrected refraction was proposed by Renaud et al. [14] using a single-element excitation, full-array waveform capture, and an adapted Kirchhoff migration developed by seismologists to image the earth subsurface. The method was validated in vivo on two young healthy subjects. No precision or accuracy values were reported. In two separate studies, Karjalainen et al. [11,45] proposed the estimation of an apparent $Ct.Th$ from $TOF$ between periosteal and endosteal bone interface at the tibia using a constant predefined speed of sound in cortical bone of 3565 m/s. This approach fails to capture the microstructural changes in porous bone structures and changes in $Ct.v_{11}$. In contrast, our method estimates $Ct.Th$ and $Ct.v_{11}$ independently; however, in this study, a very simple pore structure was assumed. Further studies should therefore target bone models with more realistic pore diameter distributions.

Recently, Iori et al. proposed a cortical backscatter model to retrieve the intracortical pore size distribution non-invasively in the tibia midshaft [46]. These findings were further supported by another study on the same set of bones, which suggested that cortical thinning and backscatter parameters describing the presence and accumulation of large

cortical pores in the tibia provide similar or better predictions of proximal femur stiffness and ultimate force than *aBMD* [20]. The cortical backscatter (CortBS) method has been applied for the first time in vivo by Armbrecht et al. [15] on postmenopausal women with low bone mineral density. The study reported a better discrimination performance for vertebral and non-vertebral fragility factures using cortical backscatter parameters (0.69 ≤ AUC ≤ 0.73) compared with DXA based *aBMD* (0.54 ≤ AUC ≤ 0.55). As the CortBS and multi-focus measurement modalities can be implemented in the same device, future in-vivo studies should be performed to evaluate if such a multiparametric assessment of macro- and microstructural (i.e., *Ct.Th* and intracortical pore size, respectively) and visco-elastic (i.e., $Ct.v_{11}$ and attenuation coefficient $\alpha(f)$) cortical bone properties can improve the discrimination and risk prediction performance for distinct types of fragility fractures. The combined estimation of *Ct.Th*, $Ct.v_{11}$ and pore size distribution using nonionizing and non-invasive technique may havea high clinical potential to prevent osteoporotic fractures.

### 4.4. Limitations

Several limitations of the proposed PAC methods were observed in this study. The methods fail for bone inclination angles larger than 7° with respect to the beam axis, as well as for the bones with high porosity values (20% or more). For bone models with tilt angles larger than 7°, most backside echoes were not captured by the receiver array resulting in a much smaller DAS beamformed signal and the transition from convex to concave shape of the backside signal wavefront disappeared. Subsequently, the zero-crossing point could not be retrieved (Figure A4c in Appendix D). As the bone surface inclination in the imaging plane can be reliably reconstructed, the application of the PAC methods can be easily restricted to locations, in which the surface inclination is within ±7°. Second, the simulation study was restricted to one scan position for one multi-focus measurement, while the ex vivo measurement performed the multi-focus measurement at 128 scan positions along the lateral distance; therefore, future in silico studies should simulate multi-focus measurements with more scan positions along the lateral distance and include simulation models with real bone curvature, tilt, and porosity. Moreover, compound imaging with beam steering [15,46] should be used to ensure that the bone area of interest is probed with sufficiently small beam inclinations. Third, for high porosity values, large amounts of scattering of ultrasound waves resulted in a strong attenuation and distortion of the backside signal, yielding an imprecise estimation of the confocal backside position (Figure A5c in Appendix D).

Another limitation is the use of simplified bone models. For in vivo transition, the effect of heterogeneous cortical pores and heterogeneous backside surface on the phase aberration should be investigated. Cortical pores lead to increased scattering, and therefore, increased phase aberration, which could be corrected with cross correlation analysis. Furthermore, the effect of changes regarding the speed of sound in soft tissue should be considered in the future, based on realistic simulation models. Conventional image reconstruction assumes an invariant speed of sound of 1540 m/s. Although the higher and variable velocity in bone was considered, soft tissue velocities can also vary by up to 10% between subjects depending on the relative distribution of skin, fat, and connective tissue along the bone length [16]. This leads to additional wave distortion, defocusing of bone regions, and misalignments of beamformed signals. Anderson et al. [47] showed on a tissue-mimicking phantom that a speed of sound error up to ±8% degrades the lateral resolution of the image by up to a factor of three. The mismatch between the assumed and actual speed of sound could be compensated for by evaluating the focus quality using the coherence factor proposed by Hasegawa et al. [48] or by using the minimum average sum of absolute differences between all pre-beamformed radio frequency channel data proposed by [49]. Renaud et al. proposed an autofocused method to estimate the optimal speed of sound of the overlaying soft tissue [50]. Additional phase aberration corrections by tissue structure may improve lateral resolution, signal quality and the accuracy and

precision of the measured time-of-flight through the bone for in vivo transition measurements.

Another limitation of this study is the non-repeated measurement design. Only one simulation was performed for each simulation model and the ex vivo measurement was performed once; therefore, for the reproducibility and precision of the multi-focus method for realistic bone simulations, ex vivo and in vivo measurements should be investigated in the future, by repeating the measurements by repositioning of the transducer between each measurement.

## 5. Conclusions

This study demonstrates the assessment of cortical thickness and speed of sound in the radial direction using refraction- and phase-aberration corrected MF imaging. Conventional DAS beamforming was improved using phase aberration correction methods to account for bone curvature, bone tilt, and bone material homogeneities from cortical pores. The method was developed and validated using in silico simplified bone models with and without pores, and one ex vivo measurement was performed on a human tibia cadaver. For a reliable in vivo estimation of cortical thickness and speed of sound values, the real bone structures and soft tissue velocity inhomogeneity must be considered. The derived parameters showed an improvement in precision and accuracy using phase aberration corrections and demonstrated good agreement with reference values.

## 6. Patent

K.R. has the patent "CortBS: Ultrasonic method for determining pore dimensions in cortical bone" pending.

**Author Contributions:** Conceptualization, H.N.M. and K.R.; methodology, H.N.M. and K.R.; software, H.N.M. and K.R.; validation, H.N.M. and K.R.; formal analysis, H.N.M.; investigation, H.N.M.; resources, H.N.M.; data curation, H.N.M.; writing—original draft preparation, H.N.M.; writing—review and editing, H.N.M., K.R. and M.M.; visualization, H.N.M.; supervision, K.R. and M.M.; project administration, K.R.; funding acquisition, K.R. All authors have read and agreed to the published version of the manuscript.

**Funding:** This work was supported by the German Ministry of Science and Education (BMBF KMUi grant no. 13GW0234) and by the German Ministry of Economic Affairs and Energy (BMWi grant no. 03THW08H01). The HR-pQCT was funded by Deutsche Forschungsgemeinschaft (DFG, German Research Foundation) in the framework of the "Major Research Instrumentation" funding program as defined in Art. 91b of the Basic Law, application no. INST 335/555-1. H.N.M. received a postgraduate scholarship from Charité-Universitätsmedizin Berlin.

**Institutional Review Board Statement:** The study was conducted in accordance with the Declaration of Helsinki. Ethical review and approval were not applicable for this study, as no measurements on humans have been conducted. The human bone sample was collected by the institute of Anatomy, University of Lübeck, Germany, in accordance with the German law "Gesetz über das Leichen-, Bestattungs- und Friedhofswesen des Landes Schleswig-Holstein II Abschnitt, §9 Leichen-öffnung, anatomisch", from 2 April 2005.

**Informed Consent Statement:** Not applicable.

**Data Availability Statement:** The raw data supporting the conclusions of this article will be made available by the authors, without undue reservation.

**Acknowledgments:** We gratefully thank Jennifer Hartwigs for the support in proof reading.

**Conflicts of Interest:** K.R. is the inventor on the patent applications (EP3641657A1, US 2020/0129140, CN110769754A, and JP 2019-570514) describing the multi-focus technology. The other authors declare no conflict of interest. The funders had no role in the design of the study; in the collection, analyses, or interpretation of data; in the writing of the manuscript, or in the decision to publish the results.

## Appendix A. Effect of Aperture and Semi-Aperture Angle $\theta$

The multi-focus (MF) method was introduced in [19] using a 32 element transducer for in silico validation. The bone plates were placed 4 mm below the linear array transducer. In vivo ultrasound measurements on postmenopausal women demonstrated in the study of Armbrecht et al. [15] showed larger bone to transducer ranges up to 30 mm; therefore, simulation models in this study were performed for a realistic transducer/bone distance of 15 mm. To study the effect of the aperture size on the estimations of cortical thickness (*Ct.Th*) and cortical speed of sound (*Ct.$v_{11}$*), simulation models were created with different transducer array sizes (varying from 32 to 72 elements in increments of 4 elements). For all models a flat bone plate was placed 15 mm below the transducer.

The tracked backside echo amplitudes for the bone plate model with aperture from 32- to 72-element are shown in Figure A1.

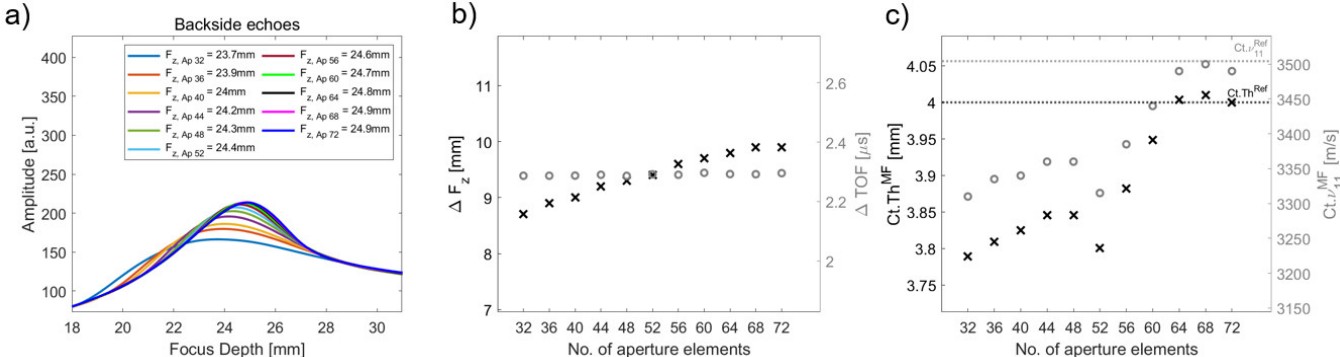

**Figure A1.** (**a**) Backside echoes simulated with different aperture sizes of a 4-mm flat bone plate model versus focus depth. (**b**) Confocal focus shift $\Delta F_z$ (black crosses) and shift in time-of-flight $\Delta TOF$ (grey circles) between the peak positions of FB echoes versus aperture size (number of aperture elements). (**c**) Estimated *Ct.Th$^{MF}$* (black crosses) and *Ct.$v_{11}^{MF}$* (grey circles) compared to the reference values *Ct.Th$^{Ref}$* (dashed black line) and *Ct.$v_{11}^{Ref}$* (dashed grey line) with respect to number of aperture elements.

The peak of the frontside echo occurred for all models at a focal depth of 15 mm. In contrast, the peak position varied for each aperture size and increased from 23.7 mm for the 32-element aperture to 24.9 mm for the 72-element aperture (Figure A1a). Moreover, the tracked front and backside (FB) amplitudes increased with an increasing aperture element number because more receiving signals were captured for delay and sum beamforming. For frontside and backside echoes, the tracked FB echo amplitudes showed a sharpening of the backside peaks with increasing aperture element number. Figure A1b shows an increase of the confocal focus shift $\Delta F_z$ with increasing aperture element number, but $\Delta TOF$ remained unchanged. The comparison of the estimated *Ct.Th$^{MF}$* and *Ct.$v_{11}^{MF}$* to the reference values in Figure A1c shows that the reference values were reached, both for *Ct.Th$^{MF}$* and *Ct.$v_{11}^{MF}$* for 64-, 68-, and 72-element apertures.

Table A1 summarizes the estimated $\Delta TOF$ between confocal FB reflection echoes, semi-aperture angle $\theta$, the critical angle $\theta_{crit}$ defined by Snell's law, the effective aperture $k_{eff}\theta$, and *Ct.Th$^{MF}$* and *Ct.$v_{11}^{MF}$*. For apertures larger than 44 elements, the difference of the semi-aperture angle to the critical angle $\Delta\theta = \theta_{crit} - \theta$ was smaller than 10° and the effective aperture was derived iteratively using [19]. For apertures less than or equal to 44-elements, no effective aperture was derived due to $\Delta\theta$ being larger than 10. In summary, the comparison of the bone plate model with different aperture element numbers revealed a dependence of the estimated *Ct.Th$^{MF}$* and *Ct.$v_{11}^{MF}$* on the semi-aperture angle. The previous study determined the effective aperture $k_{eff}\theta$ with $k_{eff} = 0.1\cdot\Delta\theta$ for $\Delta\theta < 10°$ in five iteration steps (Equation (5) in [19]). Due to the larger element number and distance of the transducer to the bone surface compared with the previous study, an adapted factor of 0.122 was used instead 0.1 for $k_{eff}$. For $k_{eff} < 0.6$, the iteration resulted in incorrect *Ct.Th$^{MF}$* and *Ct.$v_{11}^{MF}$* values; therefore, the factor $k_{eff}$ was not determined in five iterations as the

iterative process was interrupted when $k_{eff}$ reached values smaller than 0.6. For simulation models with 64-, 68-, and 72 elements the RE of $Ct.Th^{MF}$ and $Ct.v_{11}^{MF}$ was smaller than 0.5%. As simulation models with an aperture size greater than or equal to 64-elements showed no difference in $Ct.Th^{MF}$ and $Ct.v_{11}^{MF}$, all further simulations were performed with a 64-element aperture transducer.

**Table A1.** Results of $Ct.Th^{MF}$, $Ct.v_{11}^{MF}$ and relative errors (RE) for the bone plate models with different element apertures using shift in time-of-flight between confocal front-and back reflections $\Delta TOF$, semi-aperture angle $\theta_{crit}$ for the effective aperture $k_{eff}\theta$.

| Ap | $\Delta TOF$ [μs] | $\theta$ [°] | $\theta_{crit}$ [°] | $k_{eff}\theta$ [°] | $Ct.Th^{MF}$ [mm] | RE [%] | $Ct.v_{11}^{MF}$ [m/s] | RE [%] |
|---|---|---|---|---|---|---|---|---|
| 32 | 2.287 | 11.75 | 26.95 | 11.75 | 3.79 | 5.27 | 3310 | 5.54 |
| 36 | 2.287 | 13.06 | 26.73 | 13.06 | 3.81 | 4.78 | 3335 | 4.83 |
| 40 | 2.287 | 13.77 | 26.69 | 13.77 | 3.82 | 4.39 | 3340 | 4.69 |
| 44 | 2.290 | 14.96 | 26.51 | 14.96 | 3.85 | 3.87 | 3360 | 4.12 |
| 48 | 2.286 | 16.19 | 26.51 | 26.51 | 3.85 | 3.87 | 3360 | 4.12 |
| 52 | 2.291 | 17.39 | 26.90 | 20.18 | 3.80 | 4.99 | 3315 | 5.40 |
| 56 | 2.290 | 18.50 | 26.30 | 17.61 | 3.88 | 2.96 | 3385 | 3.40 |
| 60 | 2.297 | 19.65 | 25.85 | 14.97 | 3.95 | 1.30 | 3440 | 1.83 |
| 64 | 2.292 | 20.77 | 25.38 | 12.46 | 4.00 | 0.01 | 3490 | 0.41 |
| 68 | 2.292 | 21.87 | 24.92 | 13.12 | 4.01 | 0.25 | 3500 | 0.21 |
| 72 | 2.296 | 23.03 | 25.45 | 13.82 | 4.00 | 0.01 | 3490 | 0.41 |

## Appendix B. Estimation of Bone Curvature on HR-pQCT Bone Images

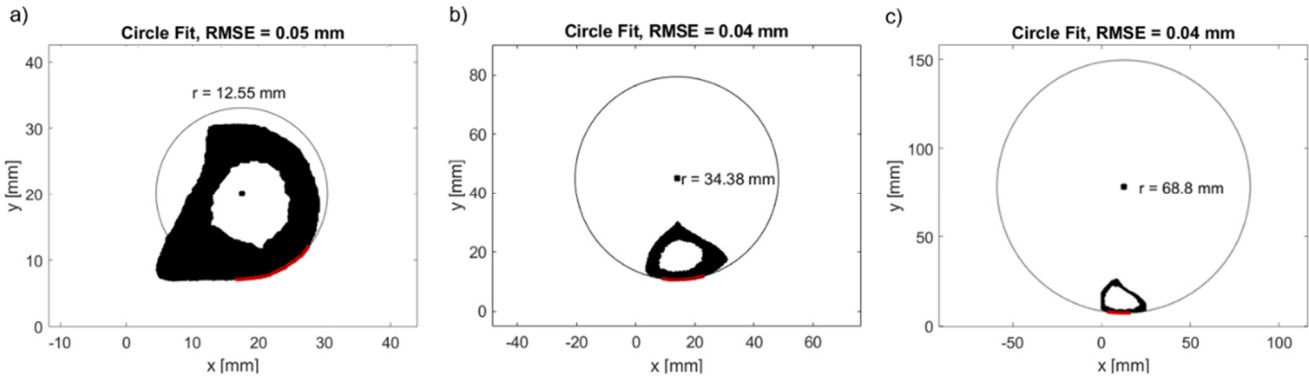

**Figure A2.** Three representative HR-pQCT scans of tibia midshaft bones of postmenopausal women. Circles were fitted to the anteromedial side to estimate the local bone surface radius. The red line indicates the central anteromedial tibia region, where ultrasound measurements were performed. The images in (**a–c**) show subjects with a minimum (12.6 mm), mean (34.38 mm), and maximum (68.8 mm) curvature radius, respectively.

## Appendix C. Phase Aberration Correction (PAC) Methods

*Appendix C.1. PAC I: Time-Shift Correction Based on Periosteal Bone Surface Geometry, Surface Time Correction (ST)*

Figure A3a shows an ultrasound wave transmitted from the transducer element 1 to the backside bone surface position F. For the curved model, the waves travel along a longer path in water (red arrows in Figure A3a) compared with the flat bone model, resulting in a shift $\Delta TOF_{geo}$ caused by the different surface geometries. These were determined using the concept of refraction compensation proposed by Yasuda et al. [24].

For each transmit channel $Tx_i$ and focus depth $F$ below the frontside surface, the crossing point of the straight ultrasound wave path and the frontside surface was determined to calculate the height length of the flat plate $h_{1,plate}$ and curved plate $h_{1,curved}$, and the width length of the flat plate $w_{1,plate}$ and curved plate $w_{1,curved}$ between the crossing point and channel position (Figure A3a). In addition, the time-of-flight from the transmitted channel to the focus point $F$ was calculated for the flat bone plate by

$$TOF_{Tx_i,plate} = \frac{\sqrt{w_{1,plate}^2 + h_{1,plate}^2}}{v_{H_2O}} + \frac{\sqrt{w_{1,plate}^2 + h_{1,plate}^2}}{Ct.v_{11}} \tag{5}$$

and for the curved bone plate using $w_{1,curved}$ and $w_{2,curved}$ instead of $w_{1,plate}$ and $w_{2,plate}$ (Figure A3a), respectively. The assumption of $Ct.v_{11}$ to calculate $TOF_{Tx_i, plate}$, was performed by implementing a loop for retrieving $Ct.Th^{MF}$ and $Ct.v_{11}^{MF}$. The starting value of $Ct.v_{11,assump}$ was defined at 2500 m/s based on the previous study [19], where $Ct.v_{11}$ values smaller than 2600 m/s were reported for cortical porosity values larger than 20%. If the difference between the calculated $Ct.v_{11}^{MF}$ and the assumed input $Ct.v_{11,assump}$ was larger than 10 m/s, the loop continued by replacing the new assumed $Ct.v_{11,assump}$ with the previously calculated $Ct.v_{11}^{MF}$. The loop stopped if the difference between calculated $Ct.v_{11}^{MF}$ and assumed $Ct.v_{11,assump}$ was smaller than 10 m/s. The total time-of-flight from one transmit channel to the receiving channel $Rx_i$ for the plate and curved models was calculated by

$$TOF_{Rx_i,plate} = TOF_{Tx_i,plate} + TOF_{64-Tx_{i+1},plate}, \tag{6}$$

under the assumption of a straight ultrasound transmitted and reflected travel paths, from transmit channel $Tx_i$ to the focus position $F$, and back to the receiving channel $Rx_i = 64 - Tx_{i+1}$.

After calculating all $TOF_{Rx}$ for all channels 1 to 64, the corrected delay is determined for each element by

$$TOF_{geo,Rx_i} = TOF_{Rx_i,curve} + TOF_{Rx_i,plate}. \tag{7}$$

In summary, *PAC I* corrects for the different propagation travel times caused by the bone curvature compared to a flat plate geometry at each receiving channel.

*Appendix C.2. PAC II: Tilt Correction (ACF)*

An autocorrelation function (ACF) analysis was used to correct for phase distortions caused by surfaces inclination. The ACF analysis was performed in the Fourier domain using the Wiener-Khinchine theorem implemented in the '*autocorr2d.m*' function [51]:

$$|V_{ACF}| = \left| F_d^{-1} \left( F_d(V_{gb}) conj \left( F_d(V_{gb}) \right) \right) \right|, \tag{8}$$

where $V_{ACF}$ is the ACF signal, $|V_{ACF}|$ the magnitude of $V_{ACF}$, and $F_d()$ and $F_d^{-1}()$ are the discrete Fourier and inverse Fourier transforms, respectively, of the gated backside signals $V_{gb}$ using Hanning-window. For each receiving channel, the backside echoes were gated after adding the beamforming delay shift, PAC I correction, and before summation (Figure A3b). From all received backside signals, the maximum signal from all received signals was used to define a threshold value for ACF correction. The threshold was defined at 40% of the maximum signal. All backside signals above the threshold were used to fit an ellipsoid on $|V_{ACF}|$ using the '*regionprops.m*' function of the Matlab Image Processing Toolbox (Figure A3c). The inclination angle of the ellipsoid to the major semi-axis $\alpha_{ACF}$ (Figure A3c) was used to apply a linear time shift correction $\Delta t_{ACF}$ to remove the tilt such that $\Delta t_{ACF}$ at the channel with the highest backside amplitude was zero. The proper correction of the wavefront tilt was verified by repeating the ACF analysis after PAC II correction (Figure A3d).

*Appendix C.3. PAC III: Cross-Correlation (CC)*

The cross-correlation method was used to correct for small fluctuations in travel times caused by intracortical pores to determine $F_{z,B}$ after ACF correction. The shift in the time-of-flight of the backside signals $\Delta TOF_{B,Rxi}$ from each receiving channel were estimated with respect to the time-of-flight of the reference channel $Rx_{Ref}$. For a focus depth smaller than the depth of the backside surface, backside echoes were not in phase. The $\Delta TOF_{B,Rxi}$ showed a concave shape with negative curvature (Figure A3e for focal depth of 23 mm and 24 mm). When focus positions converged towards confocal backside focus position $F_{z,B}$, the negative curvature of the concave shape of $\Delta TOF_{B,Rxi}$ decreased. At $F_{z,B}$ the backside signals were in phase by means of $\Delta TOF_{B,Rxi} = 0$. For a focus depth larger than $F_{z,B}$, the reflected backside signals were defocused and $\Delta TOF_{B,Rxi}$ transitioned to a convex shape with increasing positive curvature towards larger focus depths (Figure A3e for focal depths of 25 mm and 26 mm).

On the retrieved $\Delta TOF_{B,Rxi}$ a second order fit was performed to estimate the curvature parameter $p_1$ (Figure A3f) using the following equation.

$$\Delta TOF_{B,Rx_i} = p_1 \cdot \left( R_{x_i} - R_{x_{ref}} \right)^2 + p_2. \tag{9}$$

The parameter $p_2$ represents the value of $\Delta TOF_{B,Rxref}$ at the reference channel, which was not used for further analysis. The parameter $p_1$ represents the curvature of the second order fit. The change of the curvature of $\Delta TOF_{BS,Rxi}$ from negative values for focus depth smaller than the confocal focus depth towards positive values for focus depth larger than the confocal focus depth, showed a linear dependence of $p_1$ over the focus depth. The focus position where $p_1$ remained zero was defined as $F_{z,B}$ position. It was determined by a linear fit, $p_1 = m \cdot F_z + n$, from ±2 focus position around the focus depth, where $p_1$ had the smallest distance to zero (Figure A3e). Instead of using the amplitude of $H_B(F_z)$ for focus shift $\Delta F_z$ between confocal frontside and backside bone reflections, the zero-crossing value of $p_1$ (Figure A3f) was used for $F_{z,B}$ to estimate $\Delta F_z$ for *Ct.Th* and *Ct.v*$_{11}$ calculation.

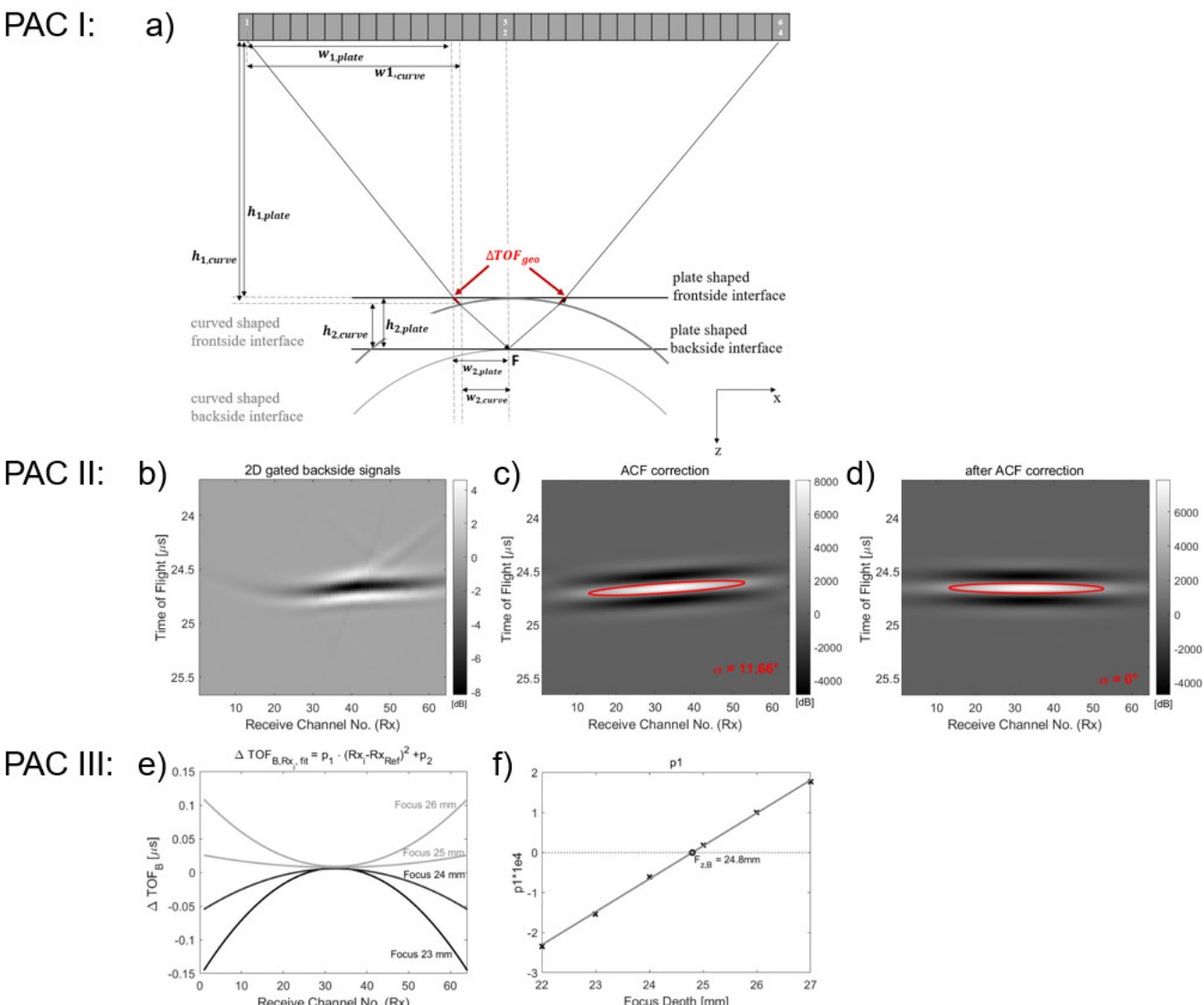

**Figure A3.** PAC I: (**a**) Schematic illustration of the additional shift $\Delta TOF_{geo}$ (red line) of a wave traveling from element 1 to focus $F$ and back to element 64 for a curved shaped bone surface compared with a flat bone plate. Note, that a focused beam of 64 elements was used and only the propagation path of the ultrasound wave of one receiving channel is shown. PAC II: Details of 2D ACF analysis for model *r40dx3.11*. (**b**) Two-dimensional image of the gated backside signals at confocal depth (25 mm) after PAC I. (**c**) Two-dimensional magnitude of ACF backside signal. The fitted ellipsoid is shown in red. (**d**) Two-dimensional magnitude of ACF backside signal after the ACF correction. PAC III: Schematic illustration to estimate $\Delta F_{z,B}$ using cross-correlation for the *flat plate* model. (**e**) Second order fit from $\Delta TOF_B$ of the backside signals using cross-correlation. (**f**) Curvature parameter of the second order fit $p_1$ (black crosses) over the focus depth and the linear fit (grey line) to estimate the zero-crossing point (black circle) for the estimation of $\Delta F_{z,B}$.

## Appendix D. Results

**Table A2.** Summary of simulation models. '*r*' and '*dx*' in the model abbreviations represent the curvature radius of the bone model and the lateral shift of the transmit and receive arrays relative to the beam axis, respectively. '*Po*' represents the porosity value when pores were simulated.

| Effect of. | Model Abbreviation | Curvature Radius *r* (mm) | Lateral Shift to Beam Axis *dx* (mm) | Bone Surface Tilt (°) | Porosity [%] |
|---|---|---|---|---|---|
| | *flat plate* | 10,000 | 0 | 0 | 0 |
| curvature | *r60dx0* | 60 | 0 | 0 | 0 |
| | *r50dx0* | 50 | 0 | 0 | 0 |
| | *r40dx0* | 40 | 0 | 0 | 0 |
| | *r30dx0* | 30 | 0 | 0 | 0 |
| | *r20dx0* | 20 | 0 | 0 | 0 |
| curvature | *r40dx1.11* | 40 | 1.11 | 1.4 | 0 |
| and | *r40dx2.11* | 40 | 2.11 | 3.1 | 0 |
| tilt | *r40dx3.11* | 40 | 3.11 | 4.5 | 0 |
| | *r40dx4.11* | 40 | 4.11 | 5.9 | 0 |
| | *r40dx5.11* | 40 | 5.11 | 7.4 | 0 |
| | *r40dx6.11* | 40 | 6.11 | 8.9 | 0 |
| curvature | *r40dx0Po2* | 40 | 0 | 0 | 2 |
| and | *r40dx0Po4* | 40 | 0 | 0 | 4 |
| porosity | *r40dx0Po6* | 40 | 0 | 0 | 6 |
| | *r40dx0Po8* | 40 | 0 | 0 | 8 |
| | *r40dx0Po10* | 40 | 0 | 0 | 10 |
| | *r40dx0Po12* | 40 | 0 | 0 | 12 |
| | *r40dx0Po14* | 40 | 0 | 0 | 14 |
| | *r40dx0Po16* | 40 | 0 | 0 | 16 |
| | *r40dx0Po18* | 40 | 0 | 0 | 18 |
| | *r40dx0Po20* | 40 | 0 | 0 | 20 |

**Table A3.** Results of $Ct.Th^{Ref}$ and $Ct.v_{11}^{Ref}$ of transmission simulation.

| Model | $Ct.v_{11}^{Ref}$ [m/s] |
|---|---|
| *r40dx0Po2* | 3428.6 |
| *r40dx0Po4* | 3321.8 |
| *r40dx0Po6* | 3189.4 |
| *r40dx0Po8* | 3127.0 |
| *r40dx0Po10* | 3038.0 |
| *r40dx0Po12* | 2953.8 |
| *r40dx0Po14* | 2848.7 |
| *r40dx0Po16* | 2774.6 |
| *r40dx0Po18* | 2704.2 |
| *r40dx0Po20* | 2681.6 |

**Table A4.** Results of $Ct.Th^{MF}$, $Ct.v_{11}^{MF}$ and relative errors (RE) for each PAC method.

| Model | Correction | $Ct.Th^{MF}$ [mm] | $RE_{Ct.Th}$ [%] | $Ct.v_{11}^{MF}$ [m/s] | $RE_{Ct.v11}$ [%] |
|---|---|---|---|---|---|
| flat plate | No | 4.00 | 0.01 | 3490 | 0.41 |
| (reference) | ST | 4.00 | 0.01 | 3490 | 0.41 |
| | ST + ACF | 4.00 | 0.01 | 3490 | 0.41 |
| | ST + ACF + CC | 4.00 | 0.01 | 3490 | 0.41 |
| *r60dx0* | No | 3.77 | 5.77 | 3330 | 5.83 |
| | ST | 3.99 | 0.32 | 3470 | 0.98 |
| | ST + ACF | 3.99 | 0.32 | 3470 | 0.98 |
| | ST + ACF + CC | 4.00 | 0.01 | 3490 | 0.41 |
| *r50dx0* | No | 3.72 | 6.91 | 3255 | 7.11 |

|  |  |  |  |  |  |
|---|---|---|---|---|---|
|  | ST | 3.98 | 0.48 | 3475 | 0.83 |
|  | ST + ACF | 3.98 | 0.48 | 3475 | 0.83 |
|  | ST + ACF + CC | 4.00 | 0.01 | 3495 | 0.26 |
| *r40dx0* | No | 3.65 | 8.70 | 3180 | 9.25 |
|  | ST | 3.96 | 1.07 | 3460 | 1.26 |
|  | ST + ACF | 3.96 | 1.07 | 3460 | 1.26 |
|  | ST + ACF + CC | 4.02 | 0.51 | 3510 | 0.16 |
| *r30dx0* | No | 3.57 | 10.80 | 3120 | 10.97 |
|  | ST | 3.94 | 1.49 | 3440 | 1.83 |
|  | ST + ACF | 3.93 | 1.65 | 3445 | 1.69 |
|  | ST + ACF + CC | 4.06 | 1.48 | 3545 | 1.16 |
| *r20dx0* | No | 3.37 | 15.66 | 2955 | 15.67 |
|  | ST | 3.82 | 4.49 | 3340 | 4.69 |
|  | ST + ACF | 3.85 | 3.78 | 3360 | 4.12 |
|  | ST + ACF + CC | 3.85 | 3.78 | 3360 | 4.12 |
| *r40dx1.11* | No | 3.67 | 8.26 | 3210 | 8.40 |
|  | ST | 3.96 | 0.91 | 3455 | 1.41 |
|  | ST + ACF | 3.96 | 1.07 | 3460 | 1.26 |
|  | ST + ACF + CC | 4.03 | 0.67 | 3505 | 0.02 |
| *r40dx2.11* | No | 3.69 | 7.66 | 3235 | 7.68 |
|  | ST | 3.93 | 1.65 | 3445 | 1.69 |
|  | ST + ACF | 3.96 | 0.91 | 3455 | 1.41 |
|  | ST + ACF + CC | 3.96 | 0.91 | 3455 | 1.41 |
| *r40dx3.11* | No | 3.67 | 7.50 | 3205 | 7.83 |
|  | ST | 3.79 | 5.23 | 3325 | 5.12 |
|  | ST + ACF | 3.95 | 1.19 | 3465 | 1.12 |
|  | ST + ACF + CC | 3.95 | 1.19 | 3465 | 1.12 |
| *r40dx4.11* | No | 3.64 | 8.99 | 3175 | 9.40 |
|  | ST | 3.58 | 10.45 | 3140 | 10.39 |
|  | ST + ACF | 3.93 | 1.86 | 3445 | 1.69 |
|  | ST + ACF + CC | 3.93 | 1.86 | 3445 | 1.69 |
| *r40dx5.11* | No | 3.61 | 9.84 | 3150 | 10.11 |
|  | ST | 3.47 | 13.28 | 3040 | 13.25 |
|  | ST + ACF | 3.80 | 4.91 | 3335 | 4.83 |
|  | ST + ACF + CC | 3.80 | 4.91 | 3335 | 4.83 |
| *r40dx6.11* | No | 3.47 | 13.37 | 3030 | 13.53 |
|  | ST | 3.29 | 17.74 | 2895 | 17.39 |
|  | ST + ACF | 3.54 | 11.48 | 3110 | 11.25 |
|  | ST + ACF + CC | 3.54 | 11.48 | 3110 | 11.25 |
| *r40dx0Po2* | No | 3.60 | 10.02 | 3065 | 10.60 |
|  | ST | 3.88 | 3.08 | 3310 | 3.46 |
|  | ST + ACF | 3.90 | 2.45 | 3330 | 2.88 |
|  | ST + ACF + CC | 3.93 | 1.83 | 3350 | 2.29 |
| *r40dx0Po4* | No | 4.39 | 9.67 | 3640 | 9.58 |
|  | ST | 3.82 | 4.57 | 3165 | 4.72 |
|  | ST + ACF | 3.81 | 4.47 | 3170 | 4.57 |
|  | ST + ACF + CC | 3.76 | 6.08 | 3135 | 5.62 |
| *r40dx0Po6* | No | 3.57 | 10.79 | 2830 | 11.27 |
|  | ST | 3.85 | 3.79 | 3050 | 4.37 |
|  | ST + ACF | 3.85 | 3.87 | 3055 | 4.21 |
|  | ST + ACF + CC | 3.92 | 1.92 | 3120 | 2.18 |

| | | | | | |
|---|---|---|---|---|---|
| *r40dx0Po8* | No | 3.54 | 11.49 | 2805 | 10.30 |
| | ST | 3.79 | 5.27 | 3015 | 3.58 |
| | ST + ACF | 3.83 | 4.18 | 3010 | 3.74 |
| | ST + ACF + CC | 3.89 | 2.87 | 3055 | 2.30 |
| *r40dx0Po10* | No | 5.16 | 29.12 | 3950 | 30.02 |
| | ST | 3.98 | 0.48 | 3030 | 0.26 |
| | ST + ACF | 3.95 | 1.24 | 3015 | 0.76 |
| | ST + ACF + CC | 3.98 | 0.48 | 3030 | 0.26 |
| *r40dx0Po12* | No | 5.58 | 39.69 | 4090 | 38.47 |
| | ST | 3.86 | 3.46 | 2835 | 4.02 |
| | ST + ACF | 3.87 | 3.23 | 2830 | 4.19 |
| | ST + ACF + CC | 3.92 | 1.94 | 2870 | 2.84 |
| *r40dx0Po14* | No | 4.78 | 19.54 | 3440 | 20.76 |
| | ST | 3.78 | 5.40 | 2760 | 3.11 |
| | ST + ACF | 3.78 | 5.40 | 2760 | 3.11 |
| | ST + ACF + CC | 3.78 | 5.40 | 2760 | 3.11 |
| *r40dx0Po16* | No | 4.82 | 20.55 | 3870 | 39.48 |
| | ST | 3.79 | 5.36 | 2635 | 5.03 |
| | ST + ACF | 3.79 | 5.14 | 2630 | 5.21 |
| | ST + ACF + CC | 3.90 | 2.58 | 2695 | 2.87 |
| *r40dx0Po18* | No | 3.82 | 4.52 | 2585 | 4.41 |
| | ST | 3.98 | 0.56 | 2680 | 0.89 |
| | ST + ACF | 3.97 | 0.79 | 2685 | 0.71 |
| | ST + ACF + CC | 3.83 | 4.21 | 2640 | 2.37 |
| *r40dx0Po20* | No | 5.65 | 41.24 | 3750 | 39.84 |
| | ST | 2.22 | 44.45 | 2675 | 0.25 |
| | ST + ACF | 3.33 | 16.66 | 2730 | 1.80 |
| | ST + ACF + CC | 3.03 | 24.14 | 2670 | 0.43 |

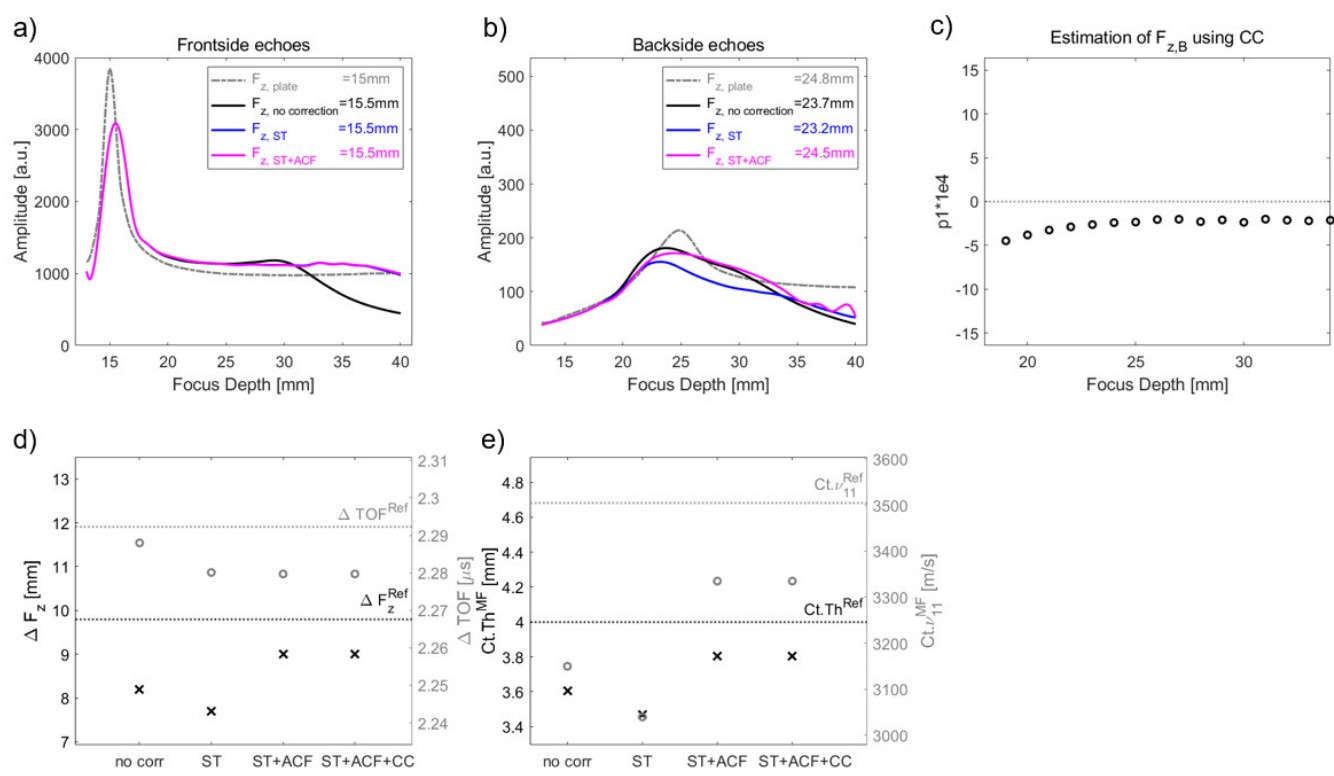

**Figure A4.** Model *r40dx5.11*: (**a**) Comparison of tracked amplitude at each correction step for frontside amplitudes (tracked amplitude after ST, ST + ACF and ST + ACF + CC correction overlap). The reference-tracked amplitude of the plate model was shown by the grey dashed line (**b**) and backside amplitudes (tracked amplitude after ST and ST + ACF correction overlap), respectively. (**c**) Curvature parameter $p_1$, retrieved from second order fit of using CC, as a function of focal depth (black circles). Linear fit (red line) was used to retrieve $F_{z,B}$ at zero-crossing point. (**d**) Comparison of focus shift $\Delta F_z$ and shift (black crosses) in time-of-flight $\Delta TOF$ (grey circles) to the reference $\Delta F_z^{Ref}$ (dashed black line) and $\Delta TOF^{Ref}$ (dashed grey line) of the plate model. (**e**) Estimated $Ct.Th^{MF}$ and $Ct.v_{11}^{MF}$ after each correction step compared to the reference $Ct.Th^{Ref}$ (dashed black line) and $Ct.v_{11}^{Ref}$ (dashed grey line) value.

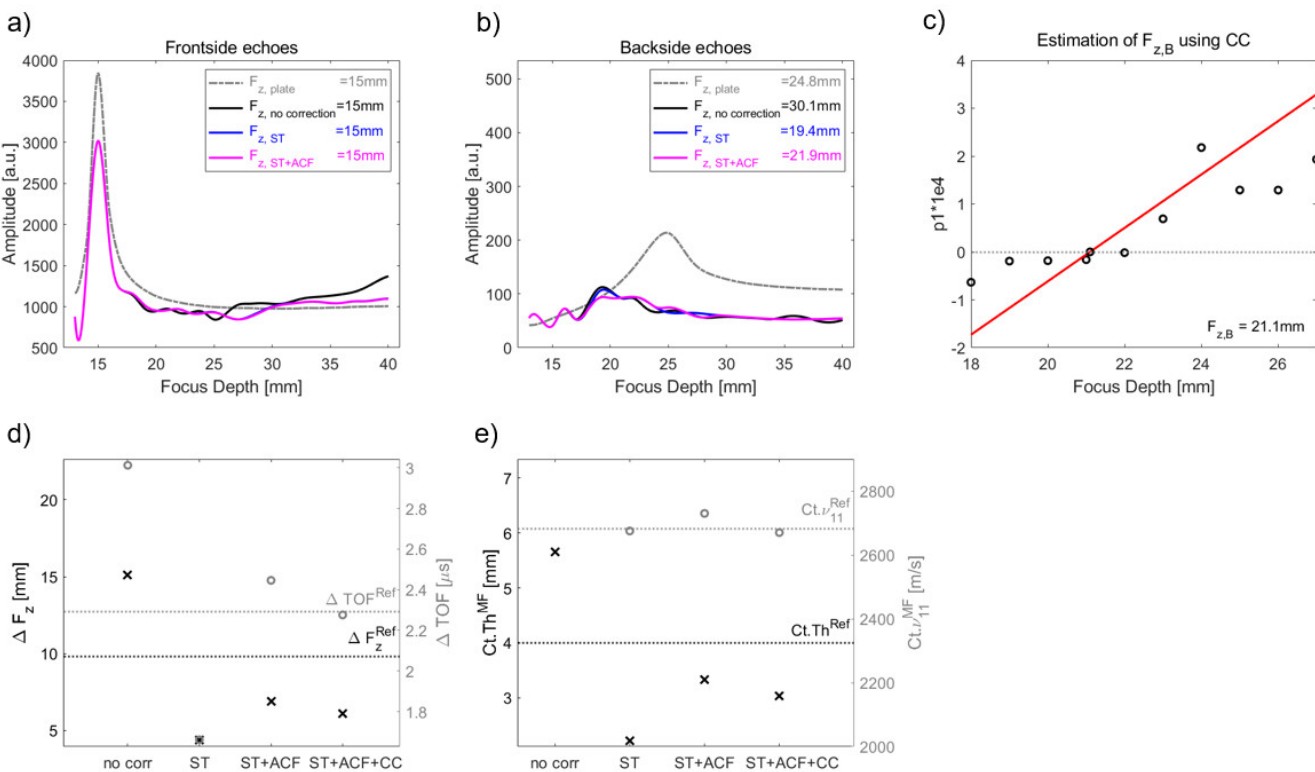

**Figure A5.** Model *r40dx0Po20*: (**a**) Comparison of tracked amplitude at each correction step for frontside amplitudes (tracked amplitude after ST, ST + ACF and ST + ACF + CC correction overlap). The reference-tracked amplitude of the plate model was shown by the grey dashed line (**b**) and backside amplitudes (tracked amplitude after ST and ST + ACF correction overlap), respectively. (**c**) Curvature parameter $p_1$, retrieved from second order fit of using CC, as a function of focal depth (black circles). Linear fit (red line) was used to retrieve $F_{z,B}$ at zero-crossing point. (**d**) Comparison of focus shift $\Delta F_z$ and shift (black crosses) in time-of-flight $\Delta TOF$ (grey circles) to the reference $\Delta F_z^{Ref}$ (dashed black line) and $\Delta TOF^{Ref}$ (dashed grey line) of the plate model. (**e**) Estimated $Ct.Th^{MF}$ and $Ct.v_{11}^{MF}$ after each correction step compared to the reference $Ct.Th^{Ref}$ (dashed black line) and $Ct.v_{11}^{Ref}$ (dashed grey line) value.

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
