# Peer review of "Estimation of Thickness and Speed of Sound for Transverse Cortical Bone Imaging Using Phase Aberration Correction Methods: An In Silico and Ex Vivo Validation Study"

_applsci, doi:10.3390/app12105283_

Round 1

Reviewer 1 Report

This study proposed a strategy for multifocus approach by correcting phase aberration caused by bone geometry and intracortical pores. The proposed method was validated by numerical simulation and experiments on a human tibia bone. The results indicated that the proposed method including phase aberration corrections provides local estimations of both cortical thickness and sound velocity and may have high clinical potential in preventing osteoporotic fractures since they were proposed as biomarkers for cortical bone quality. Overall speaking, this manuscript is well written and of scientific merit; the quality is very good as well. The experimental design and scientific writing are very rigorous, and much information and details were provided in this study to expand the domain knowledge in the field of acoustics and medical ultrasound. This manuscript was recommended for acceptance after two suggestions are addressed for minor revision.

  1. Limitations subsection should be placed at the end of Discussion.
  2. The authors could discuss more how to use the proposed method for future clinical applications in the assessment of osteoporosis.
  3. Is that possible to confirm (or discuss) the reproducibility of the proposed method? The sample size may be insufficient at the current stage.

Author Response

Thank you for the feedback. Please find enclosed also our response to the suggestions.

Reviewer 2 Report

Manuscript titled, "Estimation of thickness and speed of sound for transverse cortical bone imaging using phase aberration correction methods: an in silico and ex vivo validation study" explored the effect of phase aberration caused by a) bone curvature, b) bone tilt with respect to the beam axis, and c) material inhomogeneities due to the presence of cortical pores on the estimations of Ct.Th and Ct.ν11. The results reported in the study demonstrate the assessment of cortical thickness and speed of sound in the radial direction using refraction- and phase-aberration corrected MF imaging. Conventional DAS beamforming was improved using phase aberration correction methods to account for bone curvature, bone tilt, and bone material homogeneities from cortical pores. The manuscript is written well and the results reported in the study support the conclusions made in the study.

Author Response

Thank you for the feedback.

Reviewer 3 Report

Manuscript is well written,  regarding the line number 163 and 164 is mentioned the curvature radius r is 10.000 mm this is bit confusing does the author wants to state it's 10mm /10.00 mm or 10,000 mm which is not clear and doesn't justify the next sentences mentioning about 60mm 50mm etc

Author Response

Thank you for this comment. We changed the 10.000 mm to 10 m.

Reviewer 4 Report

In this manuscript, the authors developed multi-focus approach to correct phase aberration of cortical bone caused by its geometry and intra-pores, and provided local estimations of both cortical thickness and sound velocity, the important biomarker for cortical bone quality. The application of the three PAC methods makes sense to me, and dealing with the cortical bone ex vivo will raise additional interest to a broader audience. It is indeed an interesting piece of work. The structure of the manuscript is clear and well organized and I believe it was reasonably presented and well-cited with the relevant publications.

I have a couple minor observations:

1) According to Table A2, the 3th group of the simulation models are with curvature and tilt, but without pores. While in Table 3&4, I am confused how the authors exclude the model with porosity 20% in line “Curved tilt bone plate”, since ** was found, e.g. 4.3%(4.2%)**.

2) In Table A2, the 4th group of the simulation models are with the characters of curvature and porosity, but without tilt. I would like to whether it is possible to learn the performance of the proposed PAC method on the simulation model by adding tilt, the results may provide some reference for the researchers who execute the ultrasound experiments.

3) Please check minor typos: e.g. “2)” was omitted in line 274, when presenting the second PAC method, and the porosity value in Table A2 was also omitted.

Author Response

Thank you for the feedback.

  1. According to Table A2, the 3thgroup of the simulation models are with curvature and tilt, but without pores. While in Table 3&4, I am confused how the authors exclude the model with porosity 20% in line “Curved tilt bone plate”, since ** was found, e.g. 4.3%(4.2%)**.                          Response: Thank you for the comment. We made the mistake by using “**” for the curved tilt bone plate models and changed it to “*” because only models with tilt and without porosity were considered for that group.
  2.  In Table A2, the 4thgroup of the simulation models are with the characters of curvature and porosity, but without tilt. I would like to whether it is possible to learn the performance of the proposed PAC method on the simulation model by adding tilt, the results may provide some reference for the researchers who execute the ultrasound experiments.        Response: Thank you very much for this suggestion. There are of course this and many other aspects which are worth being investigated. However, the number of simulations and parameters investigated in this study is already large. With respect to the readability of the manuscript, we decided to add this point to the limitations in section 4.3 and will consider this for future studies.
  3. Please check minor typos: e.g. “2)” was omitted in line 274, when presenting the second PAC method, and the porosity value in Table A2 was also omitted.                                                                                             Response: Thank you for the comment. The “2)” was added in Line 285. The porosity values in A2 were also added.
